# GAME THEORETIC MIXED EXPERTS FOR COMBINATIONAL ADVERSARIAL MACHINE LEARNING

## ABSTRACT

Recent advances in adversarial machine learning have shown that defenses considered to be robust are actually susceptible to adversarial attacks which are specifically tailored to target their weaknesses. These defenses include Barrage of Random Transforms (BaRT), Friendly Adversarial Training (FAT), Trash is Treasure (TiT) and ensemble models made up of Vision Transformers (ViTs), Big Transfer models and Spiking Neural Networks (SNNs). A natural question arises: how can one best leverage a combination of adversarial defenses to thwart such attacks? In this paper, we provide a game-theoretic framework for ensemble adversarial attacks and defenses which answers this question. In addition to our framework we produce the first adversarial defense transferability study to further motivate a need for combinational defenses utilizing a diverse set of defense architectures. Our framework is called Game theoretic Mixed Experts (GaME) and is designed to find the Mixed-Nash strategy for a defender when facing an attacker employing compositional adversarial attacks. We show that this framework creates an ensemble of defenses with greater robustness than multiple state-of-the-art, single-model defenses in addition to combinational defenses with uniform probability distributions. Overall, our framework and analyses advance the field of adversarial machine learning by yielding new insights into compositional attack and defense formulations.

## 1    INTRODUCTION

Machine learning models have been shown to be vulnerable to adversarial examples Goodfellow et al. (2014); Papernot et al. (2016). Adversarial examples are inputs with small perturbations added, such that machine learning models misclassify the example with high confidence. Addressing the security risks posed by adversarial examples are critical for the safe deployment of machine learning in areas like health care Finlayson et al. (2019) and self driving vehicles Qayyum et al. (2020). However, current defenses and attacks in adversarial machine learning have trended towards a cat and mouse dynamic where in new defenses are continually being proposed and then broken Carlini & Wagner (2017); Tramer et al. (2020); Mahmood et al. (2021a); Sitawarin et al. (2022) by improved attacks.

In parallel to attack and defense development, studies have also been conducted on the transferability of adversarial examples Liu et al. (2016); Mahmood et al. (2021b); Xu et al. (2022). Transferabiltiy refers to the phenomena where adversarial examples generated for one model are also misclassified by a different machine learning model. However, to the best of our knowledge no analyses have been done on the transferability of adversarial examples designed to attack specific defenses. From these observations several pertinent questions arise:

1. *Do adversarial examples generated for one specific defense transfer to other defenses?*
2. *Based on adversarial transferability, can a game theoretic framework be developed to determine the optimal choices for both attacker and defender?*
3. *Can randomized defense selection yield higher robustness than a single state-of-the-art defense?*

These are precisely the questions our paper seeks to answer. We break from the traditional dynamic of adversarial machine learning which focuses on the single best attack and defense. We instead take

a multi-faceted approach and develop a game theoretic framework to answer the above questions. Specifically, we provide the following contributions: Most importantly, we formulate a practical, game-theoretic framework for finding the optimal strategies for an attacker and defender who each employ a set of state-of-the-art adversarial attacks and defenses. Motivated by this framework, we develop two new white-box attacks called the Momentum Iterative Method over Expectation (MIME) and the Auto Expectation Self-Attention Gradient Attack (AE-SAGA) in order to create a stronger adversary. These attacks are necessary for targeting certain randomized defenses and for adapting to multi-defense strategies. Lastly, we analyze the adversarial transferability of current defenses like Trash is Treasure Xiao & Zheng (2020), Barrage of Random Transforms Raff et al. (2019), Friendly Adversarial Training Zhang et al. (2020) and other new architectures like SNNs Rathi & Roy (2021b); Fang et al. (2021) and ViTs Dosovitskiy et al. (2020). We further leverage the low transferability between these classifiers to find those which are best suited for a combined, ensemble defense such as the one developed in our game-theoretic framework.

## 2 ADVERSARIAL MACHINE LEARNING DEFENSES

Here we summarize the state-of-the-art defenses we analyze in this paper. In the following subsections, we give an overview of each defense and our reasons for choosing said defense. It is important to note our analyses encompass a broad range of different defenses, including ones based on randomization, ones based on adversarial training and ones based on exploiting model transferability. In addition, we also consider diverse architectures including Big Transfer models (BiTs), Vision Transformers (ViTs) and Spiking Neural Networks (SNNs). Despite our broad range, we do not attempt to test every novel adversarial defense. It is simply infeasible to test every proposed adversarial machine learning defense, as new defenses are constantly being produced. However, based on our game theoretic design and open source code (which will be provided upon publication), any new defense can easily be tested and integrated into our proposed framework.

### 2.1 BARRAGE OF RANDOM TRANSFORMS

Barrage of Random Transforms (BaRT) Raff et al. (2019) utilize a set of image transformations in a random order and with randomized transformation parameters to thwart adversarial attacks. Let $t_j^i(x)$ represent the $i^{th}$ transformation used in the $j^{th}$ order in the sequence. A BaRT defense using $n$ image transformations randomly alters the input $x$:

$$t(x) = t_{\mu_n}^{\omega_n} \circ t_{\mu_{n-1}}^{\omega_{n-1}} \circ ... \circ t_{\mu_1}^{\omega_1}(x) \tag{1}$$

where $\omega$ represents the subset of $n$ transformations randomly selected from a set of $N$ total possible transformations and $\mu$ represents the randomized order in which the $n$ transformations are applied. In Equation 1 the parameters of each image transformation $t_j^i(x)$ are also randomized at run time, further adding to the stochastic nature of the defense. In this paper, we work with the original BaRT implementation which includes both differentiable and non-differentiable image transformations.

**Why we selected it:** Many defenses are broken soon after being proposed Tramer et al. (2020). BaRT is one of the few defenses that has continued to show robustness even when attacks are specifically tailored to work against it. For example, most recently BaRT achieves 29% robustness on CIFAR-10 against a customized white-box attack Sitawarin et al. (2022). It remains an open question whether using BaRT with other randomized approaches (i.e. selecting between different defenses) can yield even greater robustness.

### 2.2 FRIENDLY ADVERSARIAL TRAINING

Training classifiers to correctly recognize adversarial examples was originally proposed in Goodfellow et al. (2014) using FGSM. This concept was later expanded to include training on adversarial examples generated by PGD in Madry et al. (2018). In Zhang et al. (2020) it was shown that Friendly Adversarial Training (FAT) could achieve high clean accuracy while maintaining robustness to adversarial examples. This training was accomplished by using a modified version of PGD called PGD-$K$-$\tau$. In PGD-$K$-$\tau$, $K$ refers to the number of iterations used for PGD. The $\tau$ variable is a hyperparamter used in training which stops the PGD generation of adversarial examples earlier than the normal $K$ number of steps, if the sample is already misclassified.

**Why we selected it:** There are many different defenses that rely on adversarial training Madry et al. (2018); Zhang et al. (2019); Wang et al. (2019); Maini et al. (2020) and training and testing them all is not computationally feasible. We selected FAT for its good trade off between clean accuracy and robustness, and because we wanted to test adversarial training on both Vision Transformer and CNN models. In this regard, FAT is one of the adversarial training methods that has already been demonstrated to work across both types of architectures Mahmood et al. (2021b).

### 2.3 Trash is Treasure

One early direction in adversarial defense design was model ensembles Pang et al. (2019). However, due to the high transferability of adversarial examples between models, such defenses were shown to not be robust Tramer et al. (2020). Trash is Treasure (TiT) Xiao & Zheng (2020) is a two model defense that seeks to overcome the transferability issue by training one model $C_a(\cdot)$ on the adversarial examples from another model $C_b(\cdot)$. At run time both models are used:

$$y = C_a(\psi(x, C_b)) \tag{2}$$

where $\psi$ is an adversarial attack done on model $C_b$ with input $x$ and $C_a$ is the classifier that makes the final class label prediction on the adversarial example generated by $\psi$ with $C_b$.

**Why we selected it:** TiT is one of the newest defenses that tries to achieve robustness in a way that is fundamentally different than pure randomization strategies or direct adversarial training. In our paper, we further develop two versions of TiT. One version is based on the original proposed CNN-CNN implementation. We also test a second mixed architecture version using Big Transfer model and Vision Transformers to try and leverage the low transferability phenomena described in Mahmood et al. (2021b).

### 2.4 Novel Architectures

In addition to adversarial machine learning defenses, we also include several novel architectures that have recently achieved state-of-the-art or near state-of-the-art performance in image recognition tasks. These include the Vision Transformer (ViT) Dosovitskiy et al. (2020) and Big Transfer models (BiT) Kolesnikov et al. (2020). Both of these types of models utilize pre-training on larger datasets and fine tuning on smaller datasets to achieve high fidelity results. We also test Spiking Neural Network (SNNs) architectures. SNNs are a competitor to artificial neural networks that can be described as a linear time invariant system with a network structure that employs non-differentiable activation functions Xu et al. (2022). A major challenge in SNNs has been matching the depth and model complexity of traditional deep learning models. Two approaches have been used to overcome this challenge, the Spiking Element Wise (SEW) ResNet Fang et al. (2021) and transferring weights from existing CNN architectures to SNNs Rathi & Roy (2021a). We experiment with both approaches in our paper.

**Why we selected it:** The set of adversarial examples used to attack one type of architecture (e.g. a ViT) have shown to not be misclassified by other architecture types (e.g. a BiT or SNN) Mahmood et al. (2021b); Xu et al. (2022). While certain white-box attacks have been used to break multiple undefended models, it remains an open question if different architectures combined with different defenses can yield better performance.

## 3 Adversarial Attacks

In our paper we assume a white-box adversarial threat model. This means the attacker is aware of the set of all defenses $D$ that the defender may use for prediction. In addition, $\forall d \in D$ the attacker also knows the classifier weights $\theta_d$, architecture and any input image transformations the defense may apply. To generate adversarial examples the attacker solves the following optimization problem:

$$\max_{\delta} \sum_{d \in D} L_d(x + \delta, y; d) \quad \text{subject to: } ||\delta||_p \leq \epsilon \tag{3}$$

where $D$ is the set of all possible defenses (models) under consideration in the attack, $L_d$ is the loss function associated with defense $d \in D$, $\delta$ is the adversarial perturbation, and $(x, y)$ represents the

original input with corresponding class label. This is a more general formulation of the optimization problem allowing the attacker to attack single or multi-model classifiers. The magnitude of this perturbation $\delta$ is typically limited by a certain $l_p$ norm. In this paper we analyze the attacks and defenses using the $l_\infty$ norm.

Static white-box attacks such as the Projected Gradient Descent (PGD) Madry et al. (2018) attack often perform poorly against randomized defenses such as BaRT or TiT. In Xiao & Zheng (2020) they tested the TiT defense against an attack designed to compensate for randomness, the Expectation over Transformation attack (EOT) attack Athalye et al. (2018). However, it was shown that the EOT attack performs poorly against TiT (e.g. $20\%$ or worse attack success rate). For attacking BaRT, in Sitawarin et al. (2022) they proposed a new white-box attack to break BaRT. However, this new attack requires that the image transformations used in the BaRT defense be differentiable, which is a deviation from the original BaRT implementation.

**Attack Contributions:** It is crucial for both the attacker and defender to consider the strongest possible adversary when playing the adversarial examples game. Thus, we propose two new white-box attacks for targeting randomized defenses. The first attack is designed to work on individual randomized defenses and is called the Momentum Iterative Method over Expectation (MIME). To the best of our knowledge, MIME is the first white-box attack to achieve a high attack success rate ($> 70\%$) against TiT. MIME is also capable of achieving a high attack success rate against BaRT, even when non-differentiable transformations are implemented as part of the defense. Our second attack, is designed to generate adversarial examples that work against multiple type of defenses (both randomized and non-randomized defenses) simultaneously. This compositional attack is called, the Auto Expectation Self-Attention Gradient Attack (AE-SAGA).

### 3.1 MOMENTUM ITERATIVE METHOD OVER EXPECTATION

We develop a new attack white-box attack specifically designed to work on defenses that inherently rely on randomization, like Barrage of Random Transforms (BaRT) Raff et al. (2019) and Trash is Treasure (TiT) Xiao & Zheng (2020). Our new attack is called the Momentum Iterative Method over Expectation (MIME). The attack "mimes" the transformations of the defender in order to more precisely model the gradient of the loss function with respect to the input after the transformations are applied. To this end, MIME utilizes two effective aspects from earlier white-box attacks, momentum from the Momentum Iterative Method (MIM) Dong et al. (2018) attack and repeated sampling Athalye et al. (2018) from the Expectation Over Transformation (EOT) attack:

$$x_{adv}^{(i)} = x_{adv}^{(i-1)} + \epsilon_{\text{step}}g^{(i)} \tag{4}$$

where the attack is computed iteratively with $x_{adv}^{(0)} = x$. In Equation 4 $g^{(i)}$ is the momentum based gradient of the loss function with respect to the input at iteration $i$ and is defined as:

$$g^{(i)} := \gamma g^{(i-1)} + \mathbb{E}_{t \sim T}\left[\frac{\partial L}{\partial t(x_{adv}^{(i)})}\right] \tag{5}$$

where $\gamma$ is the momentum decay factor and $t$ is a random transformation function drawn from the defense's transformation distribution $T$. In Table 1 we show experimental results for the MIME attack on CIFAR-10 randomized defenses (TiT and BaRT). It can clearly be seen that MIME has a better attack success rate than both APGD Croce & Hein (2020) and MIM Dong et al. (2018).

| Attack | BaRT-1 | BaRT-5 | BaRT-10 | TiT (BiT/ViT) | TiT (VGG/RN) |
|---|---|---|---|---|---|
| MIME-10 | **3.18%** | 15.5% | 43.2% | 10.1% | 24.9% |
| MIME-50 | 4.3% | **8.22%** | **23.2%** | **8.3%** | **23.3%** |
| MIM | 6.7% | 39.5% | 59.5% | 52% | 58.9% |
| APGD | 8.9% | 47.7% | 70.8% | 68.2% | 40.7% |
| Clean | 98.4% | 95.3% | 92.5% | 90.1% | 76.6% |

Table 1: Performance of the MIME attack against CIFAR-10 randomized defenses: Trash is Treasure (TiT) and Barrage of Random Transforms (BaRT). In our experiments $\epsilon_{max} = 0.031$ for all attacks. It can clearly be seen that MIME outperforms both APGD and MIM on these two randomized defenses. Further defense and attack implementation details are given in Table **??**

### 3.2 AUTO EXPECTATION SELF-ATTENTION GRADIENT ATTACK

The use of multi-model attacks are necessary to achieve a high attack success rate when dealing with ensembles that contain both CNN and non-CNN model architectures like the Vision Transformer (ViT) Dosovitskiy et al. (2020) and Spiking Neural Network (SNN) Fang et al. (2021). This is because adversarial examples generated by single model white-box attacks generally do not transfer well between CNNs, ViTs and SNNs Mahmood et al. (2021b); Xu et al. (2022). In addition it is an open question if multi-model attacks can be effective against the current state-of-the-art defenses. In this paper, we expand the idea of a multi-model attack to include not only different architecture types, but also different defenses. The generalized form of the multi-model attack is found in Equation 3.

For a single input $x$ with corresponding class label $y$, an untargeted multi-model attack is considered successful if $(\forall d \in D, C_d(x + \delta) \neq y) \wedge (||\delta||_p \leq \epsilon)$. One formulation of the multi-model attack is the Auto Self-Attention Gradient Attack (Auto SAGA) which was proposed in Xu et al. (2022) to iteratively attack combinations of ViTs, SNNs and CNNs:

$$x_{adv}^{(i+1)} = x_{adv}^{(i)} + \epsilon_{step} * \text{sign}(G_{blend}(x_{adv}^{(i)})) \qquad (6)$$

where $\epsilon_{step}$ was the step size used in the attack. In the original formulation of Auto-SAGA, $G_{blend}$ was a weighted average of the gradients of each model $d \in D$. By combining gradient estimates from different models, Auto-SAGA is able to create adversarial examples that are simultaneously misclassified by multiple models. One limitation of Auto-SAGA attack is that it does not account for defenses that utilize random transformations. Motivated by this, we can integrate the previously proposed MIME attack into the gradient calculations for Auto-SAGA. We denote this new attack as the Auto Expectation Self-Attention Gradient Attack (AE-SAGA). Both SAGA and AE-SAGA use the same iterative update (Equation 6). However, AE-SAGA uses the following gradient estimator:

$$G_{blend}(x_{adv}^{(i)}) = \gamma G_{blend}(x_{adv}^{(i-1)}) + \sum_{k \in D \setminus R} \alpha_k^{(i)} \phi_k^{(i)} \odot \frac{\partial L_k}{\partial x_{adv}^{(i)}} + \sum_{r \in R} \alpha_r^{(i)} \phi_r^{(i)} \odot (\mathbb{E}_{t \sim T}[\frac{\partial L_r}{\partial t(x_{adv}^{(i)})}]) \quad (7)$$

In Equation 7 the two summations represent the gradient contributions of sets $D \setminus R$ and $R$, respectively. Here we define $R$ as the set of randomized defenses and $D$ as the set of all the defenses being targeted. In each summation $\phi$ is the self-attention map Abnar & Zuidema (2020) which is replaced with a matrix of all ones for any defense that does not use ViT models. $\alpha_k$ and $\alpha_r$ are the associated weighting factors for the gradients for each deterministic defense $k$ and randomized defense $r$, respectively. Details of how the weighting factors are derived are given in Xu et al. (2022).

## 4 TRANSFERABILITY EXPERIMENTS

Adversarial transferability refers to the phenomena in which adversarial examples generated to attack one model are also misclassified by a different model. Adversarial transferability studies have been done on a variety of machine learning models Liu et al. (2016); Mahmood et al. (2021b); Xu et al. (2022). However, to the best of our knowledge, adversarial transferability between different state-of-the-art defenses has not been conducted. This transferability property is of significant interest because a lack of transferability between different defenses may indicate a new way to improve adversarial robustnes.

In Table 2 we show the different single defenses we analyze in this paper and the best attack on each of them from the set of attacks (MIM Dong et al. (2018), APGD Croce & Hein (2020) and MIME (proposed in this work). In Figure 1 we visually show the transferability results of these attacks for CIFAR-10. We give detailed discussions of these results in our supplementary material and briefly summarize of the key takeaways from these experiments.

In general adversarial examples generated using the best attack on one defense *do not* transfer well to other defenses. For example, only $0.8\%$ of the adversarial examples generated by the BaRT-1 defense transfer to the FAT ViT defense. The average transferability for the 8 different defenses shown in Figure 1 is only $21.62\%$ and there is no single model attack that achieves strong performance (i.e. $> 50\%$) across all defenses. These results in turn motivate the development of a game theoretic framework for both the attacker and defender. For the attacker, this prompts the need to use multi-model attack like AE-SAGA that was proposed in Section 3, as no single attack (APGD, MIM or MIME) is ubiquitous. For the defender these results highlight the opportunity to increase

| Defense | Best Attack | Clean Acc | Robust Acc |
|---------|-------------|-----------|------------|
| B1 | MIME | 98.40% | 3.40% |
| B5 | MIME | 95.30% | 15.00% |
| B10 | MIME | 92.50% | 43.50% |
| RF | APGD | 81.89% | 52.00% |
| VF | APGD | 92.36% | 25.00% |
| ST | APGD | 91.54% | 0.00% |
| SB | APGD | 81.16% | 1.60% |
| BVT | MIME | 90.10% | 8.60% |
| VRT | MIME | 76.60% | 26.20% |

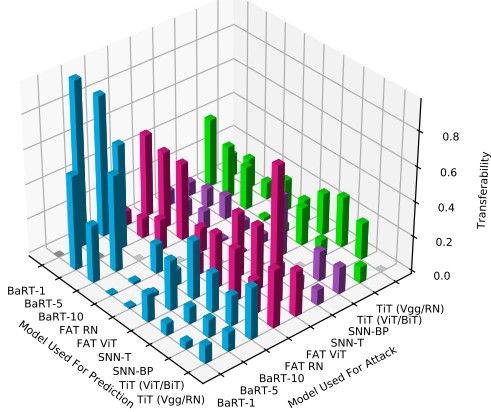

Table 2: Single defense implementations for CIFAR-10 with the corresponding strongest attack on the defense and the clean accuracy of the defense. The robust accuracy is measured using 1000 adversarial examples. The examples are class wise balanced and also correctly recognized by all the defenses in their original clean form. All attacks are done using $\epsilon_{max} = 0.031$. Complete attack and defense implementation details are given in our supplementary material.

Figure 1: Visual representation of the transferability of adversarial examples between defenses for CIFAR-10. The blue, green, pink, and purple bars represent adversarial examples generated using the best attack for BaRT, TiT, FAT, and SNN classifiers respectively. Full numerical tables used to generate this figure are given in our supplemental material.

robustness by taking advantage of the low levels of transferability between defenses through the implementation of a randomized ensemble defense.

## 5 GAME THEORETIC MIXED EXPERTS

In this section we derive our framework, Game theoretic Mixed Experts (GaME), for approximating a Nash equilibrium in the adversarial examples game. In comparison to other works Meunier et al. (2021) Pinot et al. (2020) Pal & Vidal (2020) Balcan et al. (2022) Le et al. (2022) we take a more discretized approach and solve the approximate version of the adversarial examples game. This ultimately leads to the creation of a finite, tabular, zero-sum game that can be solved in polynomial time using linear programming techniques. In relation to our work, a similar adversarial game framework was proposed in Sengupta et al. (2018) but did not include comprehensive defender and attacker threat models. Specifically, we develop our framework under an adversary that can employ state-of-the-art single model and multi-model attacks and a defender that can utilize both randomization and voting schemes.

### 5.1 THE ADVERSARIAL EXAMPLES GAME

We build upon and discretize the adversarial examples game explored in Meunier et al. (2021). The adversarial examples game is a zero-sum game played between two players: the attacker, $p_A$, and the defender $p_D$. Let $\mathcal{X}$ be the input space and $\mathcal{Y}$ the output space of $p_D$'s classifiers, and let $\Theta$ represent the space of classifier parameters. Additionally, let $P_{\epsilon,\mathcal{X}} = \{x \in \mathcal{X} : ||x||_p \leq \epsilon\}$ be the set of valid adversarial perturbations for norm $p$ and $\epsilon \in \mathbb{R}^+$.

Let $A_{\epsilon}^* = \{(f : \Theta \times \mathcal{X} \times \mathcal{Y} \to P_{\epsilon,\mathcal{X}})\}$ be the set of all valid attack functions. The goal of $p_A$ is to choose $a \in A_{\epsilon}^*$, which maximizes the expected loss of $p_D$'s classifier, $\theta$, given some pair of input and ground truth label $(x, y) \in \mathcal{X} \times \mathcal{Y}$. The goal of $p_D$ is to minimize this loss through its choice of $\theta \in \Theta$. We can thus formulate the adversarial examples game as a mini-max optimization problem:

$$\inf_{\theta \in \Theta} \sup_{a \in A_{\epsilon}^*} \mathbb{E}_{(x,y) \sim \mathcal{X} \times \mathcal{Y}}[L(x + a(\theta, x, y), y; \theta)] \tag{8}$$

Due to the vastness of $\Theta$ and $A_\epsilon^*$, solving this optimization problem directly is currently computationally intractable. To this end, in the next subsections we will formulate GaME$_1$ and GaME$_n$ which discretize $\Theta$ and $A_\epsilon^*$ by enlisting a set of state-of-the-art attacks and defenses.

## 5.2 GaME$_1$

The goal of the GaME framework is to find an approximate solution to the adversarial examples game through the implementation of a set of attacks and defenses which will serve as experts for $p_A$ and $p_D$ respectively.

Let $A' \subset A_\epsilon^*$ be a subset of all valid adversarial attack functions chosen by $p_A$. Additionally, let $D \subset \Theta$ be a set of defense classifiers chosen by $p_D$. We further impose that all $a \in A'$ are white-box attacks (see Section 3 for our adversarial threat model) and that $A', D$ are finite, i.e. $|A'| \leq N_a$ and $|D| \leq N_d$ for some $N_a, N_d \in \mathbb{N}$.

It is important to note that each $a \in A'$ is a function of some classifier, $\theta \in \Theta$, in addition to the input and ground truth label. Due to this it is possible for $p_A$ to chose to attack defense $d \in D$ with attack $a \in A'$, while $p_D$ chooses to evaluate the sample using defense $d' \in D$ where $d \neq d'$. Therefore, for convenience, we will define a new, more general set of attack strategies for $p_A$:

$$A \subseteq \{(f : \mathcal{X} \times \mathcal{Y} \to P_{\epsilon,\mathcal{X}}) : f(x,y) = a_i(U, x, y),\ a_i \in A',\ U \subseteq D\} \tag{9}$$

where we extend the definition of $A' \subseteq A_\epsilon^*$ to attack functions that can take subset of defense parameters $U \subseteq D$ as input (see Equation 3 for our multi-model attack formulation). This comes into play with multi-model attacks like AE-SAGA. Thus we will let $D$ be the strategy set of $p_D$, and $A$ be the strategy set of $p_A$. We can then reformulate a discretized version of the adversarial examples game as follows:

$$\min_{d \in D} \max_{a \in A} \mathbb{E}_{(x,y) \sim \mathcal{X} \times \mathcal{Y}}[L(x + a(x,y), y; d)] \tag{10}$$

In the above two formulations we optimize over the set of pure strategies for the attacker and defender. However, as previously explored in Araujo et al. (2020) Meunier et al. (2021), limiting ourselves to pure strategies severely inhibits the strength of both the attacker and defender. Thus we create the following mixed strategy vectors for $p_A, p_D$:

$$\lambda^A \in \{r \in [0,1]^{|A|} : ||r||_1 = 1\},\ \ \lambda^D \in \{r \in [0,1]^{|D|} : ||r||_1 = 1\} \tag{11}$$

here $\lambda^A$ and $\lambda^D$ represent the mixed strategies of $p_A$ and $p_D$ respectively. Let each $a_i \in A$ and $d_i \in D$ correspond to the $i^{th}$ elements of $\lambda^A$ and $\lambda^D$, $\lambda_i^A$ and $\lambda_i^D$, respectively. Formally:

$$\mathbb{P}(\{a_i \in A : a = a_i\}) = \lambda_i^A,\ \ \mathbb{P}(\{d_i \in D : d = d_i\}) = \lambda_i^D \tag{12}$$

where $a \in A$ and $d \in D$ are random variables. With these mixed strategy vectors we can then reformulate the adversarial examples game as a mini-max optimization problem over $p_D$'s choice of $\lambda^D$ and $p_A$'s choice of $\lambda^A$:

$$\min_{\lambda^D} \max_{\lambda^A} \mathbb{E}_{(x,y) \sim \mathcal{X} \times \mathcal{Y}}[\mathbb{E}_{(a,d) \sim A \times D}[L(x + a(x,y), y; d)]] =$$
$$\min_{\lambda^D} \max_{\lambda^A} \mathbb{E}_{(x,y) \sim \mathcal{X} \times \mathcal{Y}}[\sum_{a_i \in A} \lambda_i^A \sum_{d_i \in D} \lambda_i^D [L(x + a_i(x,y), y; d_i)]] \tag{13}$$

For continuous and or non-finite $\mathcal{X}$, $D$, and $A$ solving the above optimization problem is currently computationally intractable. Thus we can instead approximate the mini-max optimization by taking $N$ Monte-Carlo samples with respect to $(x_j, y_j) \in \mathcal{X} \times \mathcal{Y}$:

$$\min_{\lambda^D} \max_{\lambda^A} \frac{1}{N} \sum_{j=0}^{N} \sum_{a_i \in A} \lambda_i^A \sum_{d_i \in D} \lambda_i^D [L(x_j + a_i(x_j, y_j), y_j; d_i)] \tag{14}$$

For convenience we will denote $r_{d_i, a_i} = \frac{1}{N} \sum_{j=0}^{N} [L(x_j + a_i(x_j, y_j), y_j; d_i)]$. Colloquially $r_{d_i, a_i}$ represents the expected robustness of defense $d_i$ when evaluating adversarial samples generated by attack $a_i$. From a game theoretic perspective, $r_{d,a}$ is the payoff for $p_D$ when they play strategy $d$ and $p_A$ plays strategy $a$. The payoff for $p_A$ given strategies $d, a$ is $-r_{d,a}$. Our mini-max optimization problem can then be simplified to:

$$\min_{\lambda^D} \max_{\lambda^A} \sum_{a_i \in A} \lambda_i^A \sum_{d_i \in D} \lambda_i^D r_{d_i, a_i} \tag{15}$$

Using all of this we can create a finite, tabular, zero-sum game defined by the following game-frame in strategic form:

$$\langle \{p_A, p_D\}, \ (A, D), \ O, \ f \rangle \tag{16}$$

where $O = \{r_{d,a} \ \forall a \in A, \ d \in D\}$ and $f$ is a function $f : A \times D \to O$ defined by $f(d, a) = r_{d,a}$. Since this is a finite, tabular, zero-sum game, we know that it must have a Nash-Equilibrium as proven in Nash (1951). Let $R$ be the payoff matrix for $p_D$ where $R_{d,a} = r_{d,a}$. It then becomes the goal of $p_D$ to maximize their guaranteed, expected payoff. Formally, $p_D$ must solve the following optimization problem:

$$\max_{r^*; \lambda^D} r^* \quad \text{subject to } \lambda^D R \geq (r^*, \cdots r^*) \text{ and } ||\lambda^D||_1 \leq 1 \tag{17}$$

This optimization problem is a linear program, the explicit form of which we provide in the supplemental material. All linear programs have a dual problem, in this case the dual problem finds a mixed Nash strategy for $p_A$. This can be done by changing the problem to a minimization problem and transposing $R$. In the interest of space we give the explicit form of the dual problem in the supplemental material as well. These linear programs can be solved using polynomial time algorithms.

## 5.3 GAME$_n$

GaME$_n$ is a more general family of games of which GaME$_1$ is a special case. In GaME$_n$, for $n > 1$, $p_D$ can calculate their final prediction based upon the output logits of multiple $d \in D$ evaluated on the same input $x$. In order to do this, $p_D$ must also choose a function to map the output of multiple defenses to a final prediction. Formally, the strategy set of $p_D$ becomes $D = D' \times F$, where $F$ is a set of prediction functions and $D'$ is defined as follows.

$$D' \subseteq \{U : U \subseteq D, \ |U| \leq n\} \tag{18}$$

Multi-model prediction functions can increase the overall robustness of an ensemble by requiring an adversarial sample to be misclassified by multiple models simultaneously Mahmood et al. (2022). In this paper we will focus on two voting functions: the majority vote Raff et al. (2019), and the largest softmax probability vote Sitawarin et al. (2022). We will refer to these as $f^h$ and $f^s$ respectively:

$$f^h(x, U) = \arg\max_{y \in \mathcal{Y}} \sum_{d \in U} \mathbb{1}\{y = \arg\max_{j \in \mathcal{Y}} d_j(x)\} \tag{19}$$

$$f^s(x, U) = \arg\max_{y \in \mathcal{Y}} \frac{1}{|U|} \sum_{d \in U} \sigma(d(x)) \tag{20}$$

where $\mathbb{1}$ is the indicator function, $\sigma$ is the softmax function, and $d_j(x)$ represents the $j^{th}$ output logit of defense $d$ when evaluated on input $x$. Solving GaME$_n$ is the same as solving GaME$_1$, except $|D|$ will be much larger. Notationally the mini-max optimization problem in terms of $r_{d,a}$ remains the same as we have simply redefined $D$, however we can redefine $r_{d,a}$ as follows:

$$r_{(U_i, f_j), a_k} = \sum_{l=0}^{N} [L(f_j(U, (x_l + a_k(x_l, y_l))), y_l)] \tag{21}$$

where $(U_i, f_j) \in D = D' \times F$. Similarly to GaME$_1$, in GaME$_n$ $r_{(U_i, f_j), a_k}$ represents the expected robust accuracy, i.e. the payoff, for $p_D$ if they play strategy $(U_i, f_j)$ and $p_A$ plays strategy $a_k$.

## 6 EXPERIMENTAL RESULTS

For our experimental results, we test on two datasets, CIFAR-10 Krizhevsky et al. and Tiny-ImageNet Le & Yang (2015). For CIFAR-10 we solved instances of GaME$_n$ using the following defenses: BaRT-1 (B1), BaRT-5 (B5), ResNet-164-FAT (RF), ViT-L-16-FAT (VF), SNN Transfer (ST), Backprop SNN (SB), and TiT using ViT and BiT (BVT). For Tiny ImageNet we solved instances of GaME$_n$ utilizing: BaRT-1, BaRT-5, ViT-L-16-FAT, and TiT using ViT and BiT. Explicit details for our experimental setup are given in the supplementary material.

Figure 2 shows a visual representation of the performance of our GaME generated defenses when compared to single model defenses. Here, and throughout the paper, we measure the robustness

of each ensemble as the lowest robust accuracy achieved by any single attack in our study. From a game-theoretic perspective this can be seen as the minimum, guaranteed utility for the defender. Of all the defenses in our study the FAT ResNet-164 had the highest robust accuracy on CIFAR-10 at 50%. On Tiny ImageNetBaRT-5 had the highest robust accuracy at 10.62%. Compared to these defenses our GaME generated ensemble achieved 63.5% robustness on CIFAR-10 and 44.5% robustness on Tiny ImageNet, leading to a 13.5% increase and 33.88% increase in robustness on each respective dataset.

In addition to this, our ensemble is able to maintain a high level of clean accuracy in spite of optimizing for robustness in isolation. In particular, the GaME framwork is able to maintain 96.2% clean accuracy and 72.6% clean accuracy on CIFAR-10 and Tiny ImageNet respectively. This is only out performed by BaRT-1 with a clean accuracy of 98.4% on CIFAR-10 and the BiT-ViT Trash is Treasure defense with a clean accuracy of 76.97% on Tiny ImageNet. It is important to note that in both these cases the robustness of the GaME framework is higher.

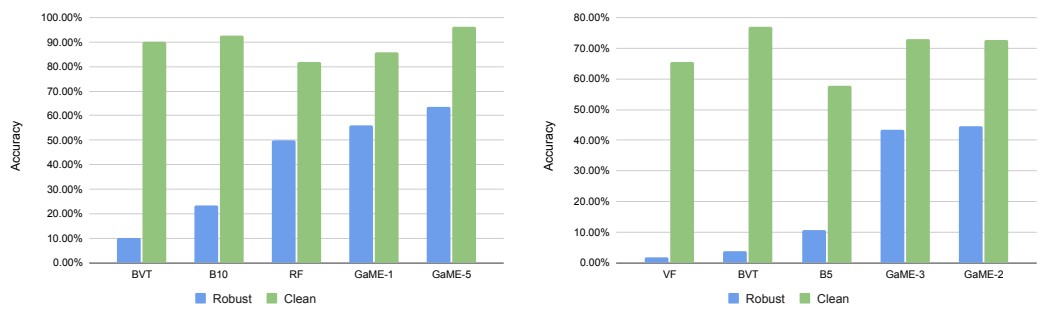

Figure 2: (Left: CIFAR-10; Right: Tiny ImageNet) Comparison of the robust and clean accuracy of the top three most robust single model defenses to the top two most robust GaME generated ensembles on CIFAR-10 and Tiny ImageNet.

We provide more detailed, numerical results for these figures in the supplementary material. Additionally, we provide studies of the effects of $n$ (the maximum ensemble size) and $N$ (the sample number) on GaME$_n$ generated ensembles along with an analysis of the computational cost of the framework.

## 7 CONCLUSION

The field of adversarial machine learning has begun to cycle between new defenses, followed by new specific attacks that break those defenses, followed by even more defenses. In this paper, we seek to go beyond this cat and mouse dynamic. We consider adversarial defense transferability, multi-model attacks and a game theoretic framework for compositional adversarial machine learning.

In terms of specific contributions, we develop two new white-box attacks, the Momentum Iterative Method over Expectation (MIME) for attacking single randomized defenses and Auto Expectation Self-Attention Gradient Attack (AE-SAGA) for dealing with a combination of randomized and non-randomized defenses. We are the first to show the transferability of adversarial examples generated by MIM, APGD, MIME and AE-SAGA on state-of-the-art defenses like FAT, BaRT and TiT.

Lastly, and most importantly, we develop a game theoretic framework for determining the optimal attack and defense strategy. Any newly proposed defense or attack can be easily integrated into our framework. Using a set of state-of-the-art attacks and defenses we demonstrate that our game theoretic framework can create a compositional defense that achieve a 13.5% increase in robustness on CIFAR-10 and a 33.88% increase in robustness on Tiny ImageNet using a multi-defense, mixed Nash strategy (as opposed to using the best single defense). Both compositional defenses for CIFAR-10 and Tiny-ImageNet also come with higher a clean accuracy than the most robust single defenses.

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
