# OpenReview forum: "Game Theoretic Mixed Experts for Combinational Adversarial Machine Learning"
_ICLR.cc/2023/Conference — Submitted to ICLR 2023_

### Official Review · Reviewer_once · 2022-10-23

**Confidence:** 4
**Correctness:** 3
**Technical Novelty And Significance:** 2
**Empirical Novelty And Significance:** 3
**Recommendation:** 5

**Clarity, Quality, Novelty And Reproducibility:**

The paper's novelty is limited in terms of technical ideas but empirically, it does evaluate a game-theoretic setup on recent defenses and attacks. Given the pace of development in this direction, it is definitely worth-while. I would definitely ask the authors to refactor their contribution statement in this way.

The lack of mention on what $N$ is and a more though study as how $N$ affects the utility estimates and thus the overall game-theoretic effectiveness, and computation cost is important to give an overall picture.

**Strength And Weaknesses:**

### Things I liked

- I like how the authors concisely described many of the recent defenses, the attacks and highlighted their relevance for this work.

- The authors propose new attacks (eg. MIME) by composing existing attacks, situate its need well in the landscape of existing attacks, and showcase its effectiveness.

### Things that need clarification / improvement

- The study of transferability of attacks on defenses, the formulation of interaction between attacks & defenses, and using game-theoretic equilibrium to come up with a mixed-strategy over defenses is not novel. [1] does all this-- given the white-box threat model, their game-theoretic formulation considers that the attacker may even know the defender's strategy and thus, considers a Stackelberg Equilibria as opposed to a Nash Eq. This paper does the study on more recent attacks and defenses and consider a more relaxed threat-model. Thus, all the 3 novel questions the paper presents on page 1 already have answers. Having said that, I do appreciate the renewed study on recent attacks and defenses but cannot support the novelty claim.

- While the reward function mostly concentrates on the loss function value for the adversarially perturbed examples, it doesn't account for the loss on the non-perturbed examples. I would surely suggest the authors to consider the loss on the latter examples in the reward function, at least for the defender, similar to [1]. Otherwise, the strategy found may over optimize for accuracy/loss on the adversarial examples while losing out on accuracy for the true test distribution.

- Given this is a zero-sum game, the Nash Eq. generates a min-max payoff and can be calculated in polynomial time. Hence the statement "These linear programs can be solved using weakly polynomial time algorithms like the interior point method Karmarkar (1984), or those developed in Vaidya (1989)" makes me wonder if the authors are aware of the subtle aspects of their formulation.

- Game_n considers a larger space (that explodes exponentially $~2^n$) of defense actions where the defender can use up to n-classifiers for ensembling. Note that this incurs much larger computational cost as one needs to estimate the utility values for each of the attack-defense pairs using $N$ samples. Hence beyond simply reporting accuracy on the adversarial examples and the test set, the authors should report the time taken to form the game-matrix. Also, the authors should discuss how effective the final strategy becomes as the number of samples used for the utility estimation ($N$), and therefore the cost, increases or decreases.

- Some study on if an attacker can attack the game-theoretic defense strategy is necessary to understand if this too is an effective defense. Based on results in [1], I would assume black-box distillation would not be as effective. Would be interesting to see how this holds for different values of $n$ $GAME_n$ and $N$ in estimating utilities.

[1] Sengupta, S., Chakraborti, T., & Kambhampati, S. (2019). Mtdeep: Moving target defense to boost the security of deep neural nets against adversarial attacks. In International Conference on Decision and Game Theory for Security.

**Summary Of The Paper:**

The paper considers a set of recent defenses against adversarial attacks on the image domain and uses existing attacks or develops them to capture a game-theoretic interaction between the attacks and defenses. To obtain the utility functions, they evaluate the effectiveness of different attacks on different defenses, thereby conducting a transferability study. Eventually, they consider a mixed strategy over a single-classifier or a less-than-n-ensemble selection for the defender by formulating the game as a zero-sum game and solving for a Nash-equilibria in this game. They show that this is more effective than using single defenses.

**Summary Of The Review:**

See above.

---

> ### Author Response · Authors · 2022-11-08
> **Response to Reviewer once (Part 1)**
>
> We appreciate the thoughtful feedback and comments. The main issue brought up in this review is about the novelty of our work in relation to [1]. This reference [1] is a very related piece of literature that we will cite in our paper. However it is important to note, our work has fundamental differences from [1]. These significant differences make our work unique, novel and worthy of publication. We elaborate further below:
>
> **Comment 1:** “The study of transferability of attacks on defenses, the formulation of interaction between attacks & defenses, and using game-theoretic equilibrium to come up with a mixed-strategy over defenses is not novel. [1] does all this-- given the white-box threat model”
>
> [1] is a great foundational work that formulates a game-theoretic approach similar to us. In our future revisions of the paper we will cite [1]. However, there are major important differences that make our paper novel. We break these differences down in terms of the attacker side, defender side and transferability.
>
> ATTACK: For the majority of datasets [1] only considers single model attacks like PGD and FSM. It is important to note, with just these single model attacks, defenses that use randomization with different architectures appear secure. However, in [Ref A] they showed that PGD fails against this kind of randomized architecture defense, while a multi-model attack like SAGA is capable of breaking the randomized architecture defense. For security game theory, if the framework does not consider multi-model attacks, the defender is given a false sense of security. In our paper, we try to specifically design the strongest multi-model attack (AE-SAGA) to avoid this issue. This is an important enhancement of [1] because it greatly improves the white-box attack success rate on a range of defenses, fundamentally changing the analysis of the game.
>
> DEFENSE: On the defender side our work also is a large expansion of what was done in [1]. In [1] they only consider a single defense type (AT), they do not consider the possibility of randomization within a defense (e.g. BaRT and TiT which our paper covers) and they only do single model output prediction. Our papers analyzed many different types of state-of-the-art defenses and architectures including BaRT, TiT, FAT, SNNs and ViTs, none of which are discussed in [1]. In addition, we consider the idea of both hard label and soft label voting for the defender, whereas [1] does not analyze voting with defenses.
>
> TRANSFERABILITY:  It is important to note that transferability is not a phenomena uniquely studied in [1], it was earlier studied in [Ref B] which we cite in our paper. In addition, the study in [1] only indirectly includes a transferability analysis of one type of defense (Adversarial Training) through its creation of the game matrix. This study in [1] is only done with FSM and PGD for the majority of the datasets. Since the publication of [1], several developments have been made including the creation of AutoAttack which outperforms PGD and the use of MIM which creates more transferable adversarial examples [Ref C]. All these attacks we analyze and use in the transferability study in our paper. Lastly, [1] only considers Adversarial Training and undefended models for transferability. This leaves an open question about the nature of transferability for randomized defenses like BaRT, Spiking Neural Networks, other types of AT (FAT) and newer defenses like Trash is Treasure. Our paper addresses this open question directly with our transferability study.
>
> **Comment 2:** “Thus, all the 3 novel questions the paper presents on page 1 already have answers.”
>
> Here we reiterate the three questions and explain why these are not answered in full by [1] or any other source to the best of our knowledge
>
>  1. ‘Do adversarial examples generated for one specific defense transfer to other defenses?’ As stated above, [1] does not study adversarial transferability with different types of defenses, does not consider newer architectures like SNNs and ViTs and does not use state-of-the-art attacks or attacks like MIM, APGD or SAGA.
> 2. ‘Can a game theoretic framework be developed to determine the optimal choices for both attacker and defender?’ Here we concede [1] has done a good job with some initial work on this question. However, this question is still unanswered when considering voting schemes and multi model predictions, different defense types other than adversarial training and most importantly, on the attack side, multi-model attacks like AE-SAGA and SAGA. All of these issues our paper addresses.
> 3. ‘Can randomized defense selection yield higher robustness than a single state-of-the-art defense?’ Again [1] answers this only for Adversarial Training defenses, doesn’t consider how to combine defense predictions like this work and doesn’t consider randomization within single defenses. Our paper expands on all of this giving a much more broad and illuminating answer.

---

> ### Author Response · Authors · 2022-11-08
> **Response to Reviewer once (Part 2)**
>
> **Comment 3:** “[1] considers a Stackelberg Equilibria as opposed to a Nash Eq”
>
> In this work we chose to consider a Nash Eq instead of a Stackelberg Eq as the order in which $p_D$ and $p_A$ choose their mixed Nash strategies largely does not affect how the game is played in a theoretical setting. A key property of the Nash Eq is that $p_A$ will be indifferent between all of their pure strategies once $p_D$ determines their mixed Nash strategy. Since we are approximating the payoff matrix, however, this will not come out perfectly in practice. Thus, we choose the attacker’s final response strategy to $p_D$’s mixed strategy to be a pure strategy where they only play an attack with the highest attack success rate. We further use this pure strategy as the metric to measure the robustness of the overall ensemble.
>
> **Comment 4:** “While the reward function mostly concentrates on the loss function value for the adversarially perturbed examples, it doesn't account for the loss on the non-perturbed examples.”
>
> The goal of our paper was to focus primarily on the robust accuracy of the defender and attack success rate of the attacker, but we do agree that clean accuracy is also important. As requested by the reviewer, an additional study on the effects of n on the GaME generated ensemble will be included in the supplemental material. These results show that the clean accuracy of a GaME generated ensemble can improve upon the clean accuracy of many of the models in the ensemble in spite of not optimizing for clean accuracy. For instance, on CIFAR-10 we are able to achieve 96.1% clean accuracy which is bested only by BaRT-1 at 98.4%. However, the ensemble greatly outperforms it in terms of robust accuracy achieving 63.5% robustness compared to 3.18%. We produce similar results for Tiny ImageNet, achieving 75.8% clean accuracy compared to 66.3% with BaRT-1. In short even without optimizing over clean accuracy, we are able to achieve high clean accuracy. Hence we leave the multi-objective optimization problem (optimizing robustness and clean accuracy) mentioned in the above comment as an exciting and distinct idea for follow up work.
>
> **Comment 5:** “Given this is a zero-sum game, the Nash Eq. generates a min-max payoff and can be calculated in polynomial time.”
> We agree and will revise the paper accordingly.
>
> **Comment 6:** “Game_n considers a larger space (that explodes exponentially $2^n$ ) of defense actions where the defender can use up to n-classifiers for ensembling… Hence beyond simply reporting accuracy on the adversarial examples and the test set, the authors should report the time taken to form the game-matrix.”
>
> The computation time for forming the game-matrix largely depends on the time needed to compute the predictions of each of the defenses for each of the attacks. We can save a large amount of time by running each set of samples, $s_i$, through each defense $d \in D$ once, receiving output $y_{i,d}$ for each sample, defense pair. Now, let $U \subseteq D$ be a subset of defenses as specified by $GaME_n$ for some n. To get the robust accuracy of $U$ when evaluating samples $s_i$ we can substitute $y_{i,d}$ for $d(s_i)$ in the computation of $F_h(s_i,U)$ or $F_s(s_i,U)$ (eq 21 and 22). This means that we do NOT need to perform a number of model evaluations that scales exponentially in n.
> The remaining computational cost comes from solving the game matrix which will scale exponentially in n. In our experiments the computational cost of this step is significantly smaller than the computational cost of evaluating each of the defenses on all of the adversarial samples. We do agree, however, that including some measure of the computation time as a function of n is important and will include one in the supplemental material.
>
> **Comment 7:** “Also, the authors should discuss how effective the final strategy becomes as the number of samples used for the utility estimation (N), and therefore the cost, increases or decreases.”
>
> We agree with this assessment and will include such a study in our supplemental material.
>
> **Comment 8:** “Some study on if an attacker can attack the game-theoretic defense strategy is necessary to understand if this too is an effective defense.”
>
> As previously mentioned, we measure the effectiveness of the defense by attacking the whole ensemble with an additional 200 samples from each of the attacks (on each model/model pairs) we used in the paper. We then used the most effective of these attacks as the final metric for the ensemble’s robustness. This shows the defense is robust even when the attacker exhaustively attacks the ensemble with all the attacks we studied and chose the best one.

---

> ### Author Response · Authors · 2022-11-08
> **Response to Reviewer once (Part 3)**
>
> **Comment 9:** “The lack of mention on what N is and a more thorough study as how N affects the utility estimates and thus the overall game-theoretic effectiveness, and computation cost is important to give an overall picture.”
>
> In the last paragraph of section 8.5 it was mentioned that we used 800 samples when computing the game matrix, we did not explicitly state N=800 however, so we will make sure this is more explicit in future revisions.
>
> **Comment 10:** “Would be interesting to see how this holds for different values of n $GaME_n$ and N in estimating utilities”
>
>  We agree with this assessment and will include a study on the effect of n on the efficacy of our GaME generated defense in the supplemental material.
>
> **Concluding Response:** We appreciate the reviewer’s feedback. We will make sure we revise our paper to show how our work both builds on [1] and also differs from [1]. We again emphasize our work has the novelty and contributions to stand on its own as a new paper. Based on this we humbly ask the reviewer to reconsider their score.
>
> References:
> [1]: Sengupta, S., Chakraborti, T., & Kambhampati, S. (2019). Mtdeep: Moving target defense to boost the security of deep neural nets against adversarial attacks. In International Conference on Decision and Game Theory for Security.
>
> [Ref A]: K. Mahmood, R. Mahmood and M. van Dijk, "On the Robustness of Vision Transformers to Adversarial Examples," 2021 IEEE/CVF International Conference on Computer Vision (ICCV), 2021, pp. 7818-7827, doi: 10.1109/ICCV48922.2021.00774.
>
> [Ref B] Liu, Yanpei, et al. "Delving into transferable adversarial examples and black-box attacks." arXiv preprint arXiv:1611.02770 (2016).
>
> [Ref C]: Mahmood, K., Gurevin, D., van Dijk, M., & Nguyen, P. H. (2021). Beware the black-box: On the robustness of recent defenses to adversarial examples. Entropy, 23(10), 1359.

---

### Official Review · Reviewer_rrfN · 2022-10-25

**Confidence:** 4
**Correctness:** 3
**Technical Novelty And Significance:** 2
**Empirical Novelty And Significance:** Not applicable
**Recommendation:** 5

**Clarity, Quality, Novelty And Reproducibility:**

- The idea of min-max problem in attack and defense is not new  (e.g. [1]) and other relevant works.
- There is no literature for other works investigating game-theoretic perspective of attacks and defenses (e.g., [2], [3], and so on).
- The writing needs a significant improvement.
- The contribution of the paper is not clear.

[1] Jingkang Wang, Tianyun Zhang, Sijia Liu, Pin-Yu Chen, Jiacen Xu, Makan Fardad, and Bo Li. Adversarial attack generation empowered by min-max optimization. Advances in Neural Information Processing Systems.

[2]  Le, T., Tuan Bui, A., Minh Tri Tue, L., Zhao, H., Montague, P., Tran, Q. &amp; Phung, D.. (2022).  On Global-view Based Defense via Adversarial Attack and Defense Risk Guaranteed Bounds . Proceedings of The 25th International Conference on Artificial Intelligence and Statistics.

[3] Cranko, Z., Menon, A., Nock, R., Ong, C. S., Shi, Z., and Walder, C. (2019). Monge blunts bayes: Hardness results for adversarial training. In Chaudhuri, K. and Salakhutdinov, R., editors, Proceedings of the 36th International Conference on Machine Learning, volume 97 of Proceedings of Machine Learning Research




**Details Of Ethics Concerns:**

There is no ethics concerns.

**Strength And Weaknesses:**

Strength
-  The paper mentions to many methods in adversarial machine learning.
-  Some experiments about the behaviors are pretty interesting though not new.

Weaknesses
-  The paper is not well-organized and well-written. The sections are not really logically linked and coherent each other.
-  Section 5 is messy with inconsistent notions.
   - $\theta(x + a(\theta, x ,y))$: $\theta$ is generally for a model parameter and it is strange to write $\theta(x + a(\theta, x ,y))$.
   - The notations $p_a$ and $a$ are confusing because it is easy to think $p_a$ depends on $a$. Similarly for $p_d$ and $d$.
   - No clear definition for $a_i(U,x,y)$. Here $U$ is a set of defenders, but why $a_i(U,x,y)$ returns a common perturbation for every defender in $U$.
-  The assumption about finite number of defenders is not feasible. We might interpret $d_i$ as a defender type (e.g., CNNs, ViTs). However, it seems not sufficiently rich to represent the model parameters.
- The main experiment in Section 6 is very humble.


**Summary Of The Paper:**

This paper considers a finite set of attackers and a finite set of defenders and views them in a game-theoretic framework to choose the optimal choices for attacker and defender.

Specifically, it solves a min-max problem to find the optimal weights for attackers and defender.




**Summary Of The Review:**

- The paper is not well-organized and well-written. The notions used are confusing.
- The theoretical setting is not feasible and practical.
- The contribution is not clear and significant.

---

> ### Author Response · Authors · 2022-11-08
> **Response to Reviewer rrfN (Part 1)**
>
> We appreciate the reviewer’s insightful comments. The main concerns of this review had to do with the organization, writing, and contributions of our paper. We address each of the reviewer's concerns in more detail below. Additionally we assert that our work is novel, presenting new results that justify a standalone paper. We particularly note that our work is unique from those cited by the reviewer [1][2][3].
>
> **Comment 1:** “Section 5 is messy with inconsistent notions.”
>
> Thank you for the suggestion, we will work to improve it.
>
> We use the notation $a_i(U,x,y)$ to denote an attack that takes multiple classifiers into consideration in its perturbation calculation. An example of such an attack is AE-SAGA as proposed in the paper. One can see a general form of the optimization problem for a multi model attack in equation 6. However, we will revise the paper to make this formulation more explicit.
>
> **Comment 2:** “The assumption about finite number of defenders is not feasible.”
>
> In this paper we are not aiming for an exact solution to the adversarial examples game, rather we are utilizing a set of defenses and attacks as “experts” in order to approximate a Nash equilibrium. We break the problem down from the most exact and explicit form to a more discrete form in section 5. Since we are approximating the equilibrium, the assertion that A and D must be finite is reasonable. In practice, and utilizing the tabular formulation we have proposed, a defender can not choose to represent an infinite number of classifiers due to memory and or time constraints. We will make this distinction more explicit in future revisions.
>
> **Comment 3:** “The main experiment in Section 6 is very humble.”
>
> We respectfully disagree. The experimental results show a strong increase in classifier robustness if one utilizes the GaME framework, leading to a 14.42% increase in robust accuracy on Tiny ImageNet over BaRT-5, and a 7% increase in robustness on CIFAR-10 over the FAT ResNet. Here the FAT ResNet is the most robust defense on CIFAR-10 and BaRT-5 is the most robust defense on Tiny ImageNet out of all of the defenses we tested. In both of these instances there is also an increase in clean accuracy: for Tiny ImageNet there is a 4.18% increase in clean accuracy over BaRT-5, and for CIFAR-10 we achieve a 2.21% increase in clean accuracy over the FAT ResNet.
>
> In addition to these results, and as suggested by reviewer “once”, we will be supplying a study on the effect of n on the robustness and clean accuracy of a GaME generated ensemble. These results show we are able to achieve a robust accuracy of 63.5% on CIFAR-10 compared to FAT ResNet, which has a robust accuracy of 50%. We are further able to improve upon the clean accuracy of all of the CIFAR-10 defenses except for BaRT-1, achieving a clean accuracy of 96.2% compared to BaRT-1’s 98.4% clean and 3.18% robust accuracy.
>
> On Tiny ImageNet we are able to produce similarly SOTA results, achieving a robust accuracy of 44.5% along with a clean accuracy of 72.6%. This outperforms BaRT-5 which has a robust accuracy 14.42% and clean accuracy 57.63%. It additionally has a higher clean accuracy than any of the single models other than The BiT-ViT Trash is Treasure defense which has 76.97% clean accuracy.
>
>  Additionally, the scope of these experiments are quite large: we attacked 7 CIFAR-10 defenses with a total of 28 different attacks and 4 Tiny ImageNet defenses with 10 different attacks totaling 38,000 adversarial samples generated and evaluated by each defense.

---

> ### Author Response · Authors · 2022-11-08
> **Response to Reviewer rrfN (Part 2)**
>
> **Comment 4:** “The idea of min-max problem in attack and defense is not new (e.g. [1]) and other relevant works.”
>
> While other works have explored mini-max optimization problems in the domain of adversarial machine learning, we believe that we are the first to study a game theoretic approach to the problem with a highly diverse set of attacks and defenses. We further note that we introduce more relaxed threat and defense models for the attacker and defender respectively. This allows the attacker to use multi-model attacker and the defender to variable subset sizes in its ensemble prediction. While [1] does use mini-max optimization, its contributions are unrelated to ours and do not take away from the novelty of our results.
>
> [1] utilizes mini-max optimization in order to create a modified PDG attack which fits certain criteria. They do not test this attack on SOTA randomized defenses like BaRT or Trash is Treasure unlike the attacks presented in our work. [1] uses their framework to design an attack that is effective against a transformation defense, an attack which maximizes the number of models fooled in an ensemble, and an attack that maximizes the number of successful adversarial samples given one model and the same adversarial noise. They do not work within a game theoretic framework, do not work with a diverse set of defenses, and do not study attack transferability. In revising our work, we will cite this work as a related attack that could be incorporated into our framework, with the explanation that [1] is not directly conflicting or overlapping with our contributions.

---

> ### Author Response · Authors · 2022-11-08
> **Response to Reviewer rrfN (Part 3)**
>
> **Comment 5:** “There is no literature for other works investigating game-theoretic perspective of attacks and defenses (e.g., [2], [3], and so on).”
>
> We disagree with the reviewer’s claim that we cited no other game-theoretic adversarial machine learning literature. The following are the main citations from this area that we have cited in our original submission:
>
> [Ref D] Laurent Meunier, Meyer Scetbon, Rafael B Pinot, Jamal Atif, and Yann Chevaleyre. Mixed nash equilibria in the adversarial examples game. In International Conference on Machine Learning, pp. 7677–7687. PMLR, 2021.
>
> [Ref E] Ambar Pal and Ren ́e Vidal. A game theoretic analysis of additive adversarial attacks and defenses. Advances in Neural Information Processing Systems, 33:1345–1355, 2020.
>
> [Ref F] Rafael Pinot, Raphael Ettedgui, Geovani Rizk, Yann Chevaleyre, and Jamal Atif. Randomization matters how to defend against strong adversarial attacks. In International Conference on Machine Learning, pp. 7717–7727. PMLR, 2020.
>
> [Ref G] Ambar Pal and Ren ́e Vidal. A game theoretic analysis of additive adversarial attacks and defenses. Advances in Neural Information Processing Systems, 33:1345–1355, 2020.
>
> The reviewer mentions [3]. Here [3] does provide interesting theoretical results which feature a mini-max optimization problem in its formulations. However, the work is not directly related to game theory and does not provide results that reduce the novelty of our work. In particular, they produce theoretical results to show that an adversary can attack a classifier irrespective of its loss, it is sufficient for one to compress an optimal transport plan between class marginals using the Lipschitz function as transportation cost, and that one can design a strong adversary utilizing a weak, black box adversary.
> 	Additionally, the theories in [3] are applied to only one dataset, namely the USPS written digits dataset and no SOTA defenses or attacks are used for comparison. This is in strong contrast to our work which implements a multitude of SOTA adversarial defenses (FAT, BaRT, and Trash is Treasure) along with multiple SOTA adversarial attacks (APGD). Our work additionally includes defense transferability studies between different defense types, and utilizes game theory to directly find an approximate Nash Equilibrium to the adversarial examples game. Thus, the results in [3] are related to adversarial machine learning as a whole, but are not strongly related to our work in particular so we believe a citation is not necessary.
>
> [2] builds upon a game theoretic framework to show that one can find upper bounds on the adversarial risk for both the defender and attacker. They leverage their theoretical work to design attack and defense algorithms which take, as the paper says, a more global perspective on the generation of adversarial samples and defending against them. This global view allows the attacker to consider not only the initial input space of the classifier, but also the latent space induced by the classifier’s intermediate layers (in the case of classifiers like CNNs, DNNs, etc.). They develop this new attack and defense in order to show how taking a global approach can improve existing SOTA defenses and attacks. In our work we also improve upon existing SOTA defenses and attacks. Our work takes a different approach, instead utilizing game theory to find an approximate Nash equilibrium to the adversarial examples game. We also propose new attacks (AE-SAGA and MIME) but formulate these algorithms with the intent of targeting certain defense types, namely random transform defenses and ensemble defenses. Our GaME framework also functions as an improvement for both the attacker and defender. We do agree that this paper provides game theoretic contributions to adversarial machine learning and is thus related to our work, therefore we will cite it in future revisions.
>
>
> **Concluding Response:** We appreciate the reviewer’s feedback. We disagree, however, with the reviewer’s assessment that our formulations are not novel and are not theoretically or practically feasible, as outlined above. We will work to ensure the consistency of our notation and cite more relevant works as the reviewer suggests. We will additionally include further studies on our results which show the effectiveness of our framework. Based on this, we kindly ask that the reviewer reconsider their score.

---

> ### Comment · Reviewer_rrfN · 2022-12-05
> **Further comments and verifications**
>
> Thanks for your feedback to my questions and comments. My main concern is the writing and technicality of Section 5 which is the main contribution of this work.
>
> First, Section 5.1 possibly misleads the readers. It is not because of the rigorousness of  Eq. (8) and its connection to the following sections. One might think that $\varTheta$ as a continuous model parameter space and you are searching the model $\theta$ that minimizes the worst-case losses by the attackers. Later, you assume a finite set of attackers in $A_\epsilon^*$ and defenders in $\varTheta$. However, Eq. (8) becomes trivial because this is only a search on a small finite numbers of pairs $(a, d)$.
>
> As far as I understand, this work considers a finite set of attackers in $A'$ and fixed defenders (i.e., classifiers) in $D$ and aims to find the optimal weighted combination of attackers and defenders. Based on $A'$, it defines $A$ as in Eq. (9). Therefore, $a \in A$ means that there exist a subset $U \subset D$ and $a' \in A'$ attacking all defenders in $U$. To use $a' \in A'$ to attack all in $U$, one can use Eq. (3), Auto-SAGA, or AE-SAGA. In worst case, the maximal number of elements in $A$ is $2^{|A'|}$.
>
> Subsequently, you endow the discrete distributions with the masses $\lambda^A$  for $A$ and $\lambda^D$ for $D$ and reach the OP in (13). Actually, I cannot see the equivalence of OP in (10) and (13). Particularly, the OP in (10) has a trivial solution. Given $d \in D$, the inner max returns the attacker $a(d) \in A$ that maximizes the loss w.r.t. $d$. By which I mean, it seems the OP in (13) is richer than that in (10). The first one has the deterministic nature, while the second one has the stochastic nature.
>
> Certainly, (13) can be solved agnatically using linear programming. Subsequently, in Game-n, you enrich D by considering the set $D'$ where each $d' \in D'$ consists of $n$ defenders in $D$. Moreover, each $d'$ is associated with an ensemble function $f^h$ or $f^s$ for ensembling the defenders in $d'$.
>
> Going back to OP in (13), it is not clear to me when running $a_i(x_j, y_j)$ on the set $U$, it is run on the elements of $D$ or $D'$? If it is run on the elements of $D'$, how you can attack the ones using voting mechanism due to its discrete nature?
>
> Another concern is about the choice of $A'$ and $D'$. For $A'$, what $U$ should be included? Similar for $D'$, what n defenders should be chosen to form an ensemble element of $D'$? Do you have a principle to guide these?
>
> To conclude, it seems to me that to improve the adversarial robustness, given a pretrained defender, your approach tries to find the optimal way to combine them?
>
> Regarding your experiments, you did not mention to the attackers used in $A'$. Additionally, for the robust accuracies in Figure 2, what attack is using? I believe that you should report the robust accuracies w.r.t. many attacks such as PGD-x, Auto-Attack, and so on, especially the ones not in $A'$ to demonstrate the generalization ability of your approach.
>
> Finally, I am keen on my current scores. This is because the contributions regarding MIME and AE-SAGA are incremental and the above reasons for the main contribution in Section 5.

---

> > ### Author Response · Authors · 2022-12-09
> > **Response to Further Comments from Reviewer rrfN (Part 1)**
> >
> > We appreciate the reviewer’s additional comments. In the following sections we explain how all the reviewer’s concerns have already been considered and accounted for in the current version of the paper or will be addressed in the final version.
> >
> > **Comment 1:** “First, Section 5.1 possibly misleads the readers. It is not because of the rigorousness of Eq. (8) and its connection to the following sections.”
> >
> > We appreciate the reviewer’s concerns, but we believe that section 5 is not misleading to the reader and clearly outlines its purpose. The main goal of section 5.1 is to show our derivation in finding an approximation to the optimization problem seen in Eq (8). We do not claim to have developed a way to directly solve (8) with a finite, tabular method. We state the following at the beginning of section 5 to make this distinction clear: “In this section we derive our framework, Game theoretic Mixed Experts (GaME), for *approximating* a Nash equilibrium in the adversarial examples game.“
> >
> > **Comment 2:** “Later, you assume a finite set of attackers in $A_\epsilon^*$ and defenders in $\Theta$. However, Eq. (8) becomes trivial because this is only a search on a small finite numbers of pairs (a,d).“
> >
> > (8) is certainly not a trivial optimization problem, however it does become easier to optimize when making the search space finite and discrete as seen in (10). This is not the main contribution of our paper, but rather a step we take to reach more rich optimization problems as seen in (13) and (21). As stated above, our framework finds an approximate mixed Nash strategy for a defender in the adversarial examples game by discretizing the search space through the implementation of multiple state-of-the-art  attacks and defenses.
> >
> > **Comment 3:** “Actually, I cannot see the equivalence of OP in (10) and (13)”
> >
> > The paper does not claim that (10) and (13) are equivalent and rather establishes (13) as an extension of (10) which allows for mixed nash strategies. This clarification was also made in the paper: “In [(8) and (10)] we optimize over the set of pure strategies for the attacker and defender. However, as previously explored in Araujo et al. (2020) Meunier et al. (2021), limiting ourselves to pure strategies severely inhibits the strength of both the attacker and defender. Thus we create the following mixed strategy vectors for pA, pD“.
> >
> > **Comment 4:** “Going back to OP in (13), it is not clear to me when running $a_i(x_j,y_j)$ on the set $U$, it is run on the elements of $D$  or $D’$?“
> >
> > First, the purpose of establishing the set $A$, of which $a_i(x_j,y_j)$ is an element, is to more explicitly define the set of all attacks that $p_A$ can run. All attacks require is for the attacker to choose some model, or models from which to base the attack’s optimization problem.Thus, we create the set $A$ to contain all the possible attacks (e.g. MIME on BaRT-1 or AE-SAGA on BaRT-5 and BVT) that $p_A$ can run given their chosen set of general attack functions (APGD, AE-SAGA). The definition of $A$ is created to be generalized, allowing the attacker to attack any subset $U \subseteq D$.
> >
> > Thus, $a_i(x_j,y_j)$ is run on the elements of $P(D)$ for any $n$ in GaME-n as $P(D)$ is the powerset of $D$. As will be covered in a later question, it will be up to the individual running the instance of GaME to determine if the whole of $P(D)$ should be used for each $a \in A’$.
> >
> > Additionally, to make it more clear, in (13) each attack $a_i(x_j,y_j) \in A$ is evaluated on a single defense at a time, this is in contrast to (21) where each attack is evaluated on $U \in D’$. We would also like to note a notational inconsistency in (21) which we have fixed for a camera-ready version: $r_{(U_i, f_j), a_k} = \sum_{l=0}^N [L(x_l + a_k(x_l,y_l), y_l ; (U_i, f_j))]$.
> >
> > **Comment 5:** ” If it is run on the elements of $D’$, how you can attack the ones using voting mechanism due to its discrete nature?”
> >
> > Every $U \in D’$ such that $|U| > 1$ will use some voting mechanism in its prediction process as multiple models are responsible for the final prediction. Whether these voting functions are considered by the attack function is dependent on the formulation of the attack function itself. Such formulations are outside the scope of this paper, but do present the possibility for interesting follow-up works.
> >
> >  In the case of AE-SAGA we target multi-model defenses by maximizing the probability that all models in the ensemble are fooled. This does not directly target any particular voting method, but is generally effective against any voting method. This is because all voting methods require the chosen voting models to predict the correct class label with top-1 or at least relatively high confidence in order to result in a correct prediction.

---

> > > ### Comment · Reviewer_rrfN · 2022-12-12
> > > **My feedback**
> > >
> > > Thanks reviewers for answering my questions. I realize that this paper has three contributions: MIME, AE-SAGA, and a game theoretic framework to optimally combine pretrained defenders for improving adversarial robustness.
> > >
> > > To me, MIME and AE-SAGA have incremental novelty because they directly combine existing techniques. For MIME, it combines MIM and BaRT. For AE-SAGA, it combines SAGA and BaRT.
> > >
> > > Regarding the last contribution, I disagree the comment: *The reviewer's comment does not accurately reflect the scope and magnitude of our work*.    As mentioned in previous post, you start from the set of base attacks in $A'$ and base defenders in $D$. You then extend $A'$ to $A$ by allowing to attack multiple defenders in $U \subset D$. In $Game_n$, you also strengthen $D$ to $D'$ by considering $n$ defenders with an ensemble function $f^h$ or $f^s$. Finally,  a linear program problem is solved to learn the weights to combine the defenders in $D'$.    This means that you learn the optimal weights to combine the fixed pre-trained defenders.
> > >
> > > There are some concerns about the last contribution. First, the writing of 5.1 and E. (8) is misleading. You should state explicitly your setting in the first place because readers might confuse you aim to update defenders. To me the definition in Eq. (9) is not rigorous although understandable with some conjecture. Specifically, in your experiments, you used AE-SAGA to attack the pairs of defenders. However, in the definition, you write $f(x,y) = a_i(U,x,y),  a_i \in A', U \subset D$. It seems that for each base attack $a_i \in A'$, you can devise an extended attack using $a_i$ to attack on the set of defenders in $U$ as shown in Eq. (3). However, Eq. (3) means that we attack uniform combination of defenders in $U$, hence I cannot see how it connects to the base attack $a_i$. Additionally, in AE-SAGA, you do not use uniform weights as in $G_{blend}$, a weighted average of the gradients of each model $d \in D$. I believe that it needs a more rigorous definition for $A$.
> > >
> > > Moreover, for CIFAR10, you employed 28 attacks in $A$ and 7 defenders. It means that for each batch, you need to run 28 attacks to work out 28 batches of adversarial examples. These will be later evaluated on 7 defenders to form the matrix of $28 \times 7$ for a linear programing solver. This is for $Game_1$. To me, it seems to be costly for one iteration with one batch. Can you please comment on this? Additionally, this is for $Game_1$ and it is unclear how you choose the elements for $D'$ in a $Game_n$. For Tiny ImageNet, you employed 10 attacks in $A$ and 7 defenders. It also requires to attack 10 times for a given batch and evaluate on 7 defenders in $Game_1$.
> > >
> > > Regarding the experiments, I have some concerns. First, according to this sentence: *throughout the paper, we measure the robustness
> > > of each ensemble as the lowest robust accuracy achieved by any single attack in our study*, it seems that you only report the lowest robust accuracy when using the base attacks in $A'$. I believe that you should testify the optimal ensemble with out-of-list attacks to demonstrate the generalization ability of a general attack. The reason is that when deployed in reality, your system can be attacked by new attacks. Can you further clarify this point? Second, one of motivation of your work is to claim that by extending $A'$ to $A$ and $D$ in $Game_1$ to $D'$ in $Game_n$, you can strengthen your game for improving adversarial robustness. The latter is partly reflected in Figure 2 for CIFAR10. However, it seems not consistent for Tiny ImageNet and I am not clear why you do not report $Game_1$ for this dataset. Moreover, there is no ablation study to demonstrate the benefit of extending $A'$ to $A$.
> > >
> > > Finally, it would be interesting if the paper conducts more experiments to compare the case of fixing the defenders and updating the defenders in the game. However, it is just my comment to further improve this work.
> > >
> > > Finally, I like the ideas of considering a game of extended attacks in $A$ and defenders in $D'$. I believe this will be a strong work if the writing of Section is improved, the mathematical formulations are formed more rigorously, and the selling points are demonstrated more adequately.  I decide to increase my score to 5.

---

> > > > ### Author Response · Authors · 2022-12-14
> > > > **Response to Further Feedback from Reviewer rrfN (Part 1)**
> > > >
> > > > We appreciate the reviewer’s efforts in engaging with us in an open discussion and providing further suggestions to improve our paper. We further appreciate the reviewer for increasing their score for our paper from a 3 to a 5. In the following sections we will address the additional feedback from the reviewer. We ask, in light of our response, for the reviewer to kindly consider increasing their score from 5 to at least 6 as we can further address their remaining comments:
> > > >
> > > > **Comment 1:** “I realize that this paper has three contributions: MIME, AE-SAGA, and a game theoretic framework to optimally combine pretrained defenders for improving adversarial robustness.”
> > > >
> > > > The paper develops MIME and AE-SAGA as more effective ways to target single-model, random-transform and ensemble defenses implementing random-transform defenses respectively. It also develops the GaME framework as a practical and effective method for improving the robustness of ensemble adversarial defenses as well as improving the effectiveness of adversarial attacks targeting such defenses. Lastly the paper contributes the first adversarial transferability study which considers the transferability between unique defense architectures. Transferability between adversarially trained defenses has been done before. However the transferability between random transform defenses, adversarially trained defenses, and spiking neural networks has never been considered before us. This is significant as it clearly shows that unlike general adversarial attacks with vanilla models (think ResNets and PGD), tailored attacks on defenses DO NOT transfer well between defenses. This opens up both new attack and defense perspectives. In summary we have new attacks, new defenses, a game theoretic framework and comprehensive experimental results, all which push the field of adversarial machine learning forward in new and exciting directions. This is the essence of novelty.
> > > >
> > > > **Comment 2:** “To me, MIME and AE-SAGA have incremental novelty because they directly combine existing techniques. For MIME, it combines MIM and BaRT. For AE-SAGA, it combines SAGA and BaRT.”
> > > >
> > > > It is important to note his claim is not correct. BaRT stands for Barrage of Random Transforms. This is a defense based on taking an input image $x$, and applying a series of $n$ transformations in a random order and with randomized parameters such that a classifier classified $T(x)$ where $T(\cdot)$ can further be defined as $t_1() \cdot t_2 \cdot … \cdot t_n$. BaRT is a defense to distort adversarial inputs using randomization. It is not an attack.
> > > >
> > > > MIM stands for Momentum Iterative Method. Simply put, this is a white-box attack that works by incorporating momentum into the update rule for a projected gradient descent style algorithm. It is not possible to combine an attack and defense as they are two fundamentally different things. Hence MIME is clearly not a combination of BaRT and MIM.
> > > >
> > > > MIME does not combine MIM and BaRT but rather MIM with Expectation Over Transformation techniques. This also holds true for AE-SAGA, combining SAGA with EOT. We believe MIME and AE-SAGA are more than incremental improvements. Their novelty comes from their effectiveness against their respective targets. While EOT attacks have been devised in the past to target defenses like BaRT, they either try to approximate the derivative w.r.t. the transformations with BPDA techniques, which was shown to be ineffective, or target a weakened version of BaRT which can only implement differentiable transformations [A].
> > > >
> > > > MIME and AE-SAGA break this trend by allowing the attacker to target any random transform defense, including BaRT defenses with non-differentiable transformations, while remaining effective. Additionally, AE-SAGA is the first ensemble attack technique designed to target multiple, unique defense architectures simultaneously: effectively attacking random transform defenses, vision transformers, and adversarially trained defenses without fine tuning of the hyperparameters.

---

> > > > ### Author Response · Authors · 2022-12-14
> > > > **Response to Further Feedback from Reviewer rrfN (Part 3)**
> > > >
> > > > **Comment 9** “I believe that you should testify the optimal ensemble with out-of-list attacks to demonstrate the generalization ability of a general attack. The reason is that when deployed in reality, your system can be attacked by new attacks. Can you further clarify this point”
> > > >
> > > > We’d be happy to further clarify this point, both here and in the revised version of our paper. Essentially from a security perspective, giving the defense access to all methods of attack would seem to be giving the defender too much of an advantage. Hence the question of out-of-list attacks. However in the field of adversarial machine learning this question has for the most part been settled in [B] which has been cited over 600 times as of writing this. In [B] they summarize the current best practices in adversarial machine learning which include using adaptive attacks as opposed to static attacks and using a variety of attacks as opposed to a large group of similar attacks. This is precisely the methodology that we adhere to in our paper. If we were to use out of list white-box attacks they would fall into one of two categories: adaptive or non-adaptive. We can consider each case to analyze why out of list attacks do not further enhance our paper or add meaningful results to the experiments.
> > > >
> > > > If we were to use non-adaptive out of list white-box attacks, this goes against the principles advocated in [B] as these are not the most effective attacks on defenses and can give a misleading sense of robustness. For example in our paper we did not experiment with FGSM. Why? This is because it is an attack similar in nature to PGD and AutoAttack but in almost all cases performs worse. If we report FGSM as the out of list attack robustness, it would give an unrealistic sense of the security of any defense as there are stronger adaptive attacks. Hence we cannot in good faith use static white-box attacks as our out of list attack.
> > > >
> > > > This brings us to our next point which is why not use out-of-list adaptive attacks? In this scenario we run into the perplexing question. Why does the defender not have knowledge of adaptive attacks? For every defense in the paper, we have some form of customized adaptive attack either proposed by us or published in the literature and it is known from [B] that this is generally the strongest type of attack and the one on which robustness should be measured. Handicapping the defender in this manner would require a defense designer that is simply ignorant of the literature (e.g. [B]) when building their defense. This is not something that can logically be considered realistic.
> > > >
> > > > In summary broadly speaking there are two types of out of list attacks (adaptive and non-adaptive). While it is tempting to use a non-adaptive attack at the out of list choice, this does not give a realistic robustness and hence we do not use it. For adaptive attacks an out of list adaptive white box attack that the defender is not privy to would represent a defender who is not capable of correctly analyzing primary security principles, a nonsensical result. It is important to understand the GaME framework is pushing the boundaries of both the attacker and the defender to their utmost capability to ascertain which one is coming out on top. This cannot be done if either attacker or defender becomes limited in their performance.
> > > >
> > > > **Comment 10** “The latter is partly reflected in Figure 2 for CIFAR10. However, it seems not consistent for Tiny ImageNet and I am not clear why you do not report $\text{GaME}_1$ for this dataset.”
> > > >
> > > > The intention of this visualization was to show the three most robust single-model defenses in comparison to the two strongest GaME generated ensembles for each dataset. We realize, now, that we included GaME-1 in the CIFAR-10 plot in error as it is not the second strongest GaME generated defense. We do see the importance of including a more direct, empirical comparison between GaME-1 and GaME-n in the paper. Thus we will replace the GaME-3 results with GaME-1 in the Tiny ImageNet plot of figure 2.
> > > >
> > > > **Comment 11** “Moreover, there is no ablation study to demonstrate the benefit of extending $A’$ to $A$.”
> > > >
> > > > Without the multi-model attacks included in A, like AE-SAGA, the attacker would perform far worse. This can be verified experimentally, if required, by rerunning all experiments without AE-SAGA as proof, although this would be superfluous. Additionally, as discussed in our response to comment 9, no competent defender would choose to omit attacks like AE-SAGA.

---

> > ### Author Response · Authors · 2022-12-09
> > **Response to Further Comments from Reviewer rrfN (Part 2)**
> >
> > **Comment 6:** ”Another concern is about the choice of $A’$ and $D’$. For $A’$, what $U$ should be included? Similar for $D’$, what n defenders should be chosen to form an ensemble element of ? Do you have a principle to guide these?”
> >
> > For $A$ this is dependent on the general attack functions chosen and is up to the design of the individual running the instance of GaME. From a game-theoretic perspective, it is best to run each attack $a \in A’$ against each subset $U \subseteq D$ in order to create the strongest possible adversary, however for practical reasons this is not always reasonable. For instance, running single model attacks like PGD against an ensemble will likely reduce the effectiveness of the attack and will be computationally expensive, thus one will likely choose to omit such an attack. Here we have strong experimental evidence in the previous literature to back up these claims. Specifically, in [A] it has been shown that PGD does not transfer well between models compared to other white-box attacks. Hence PGD is not an effective way to attack a mixed defense ensemble. We also show this in our transfer results in Section 4. A natural question then is which subset of attacks are best? In our paper we consider the best SOTA available at the time of experimentation e.g. AutoAttack, SAGA, AE-SAGA and MIME. It is important to note that as the field advances more attacks may be added to this subset for testing. This is a strength of our framework in the sense that new advances on the attack side of adversarial machine learning can be easily incorporated.
> >
> > $D’$ on the other hand always contains all subsets of $D$ up to size n, it is up to the game-matrix and its solution to determine which of these subsets should be included in the final ensemble. Additionally, in the supplementary material we include a study on the effect of $n$ on the robustness of GaME generated defenses. The study shows that the ensemble defenses increase in both robustness and clean accuracy as one increases $n$. We also provide a study in the supplementary material which analyzes the computational cost of the framework as n increases. It shows that the computational cost increases swiftly for small values of n, and plateaus for larger values of n. These two studies should give the reader a good idea of the tradeoff between defense robustness and computational cost. In general, if one has the computational resources, it is best to choose $n = |D|$.
> >
> > We do not make any generalizing claims on this topic in the paper, however, as best practice will vary widely depending on the choice of $A’$ and $D$. Thus we instead choose to provide effective and clear studies in the supplementary material and design our framework to be flexible for all choices of $A’$ and $D$.
> >
> > **Comment 7:** “To conclude, it seems to me that to improve the adversarial robustness, given a pretrained defender, your approach tries to find the optimal way to combine them?”
> >
> > The reviewer's comment does not accurately reflect the scope and magnitude of our work. Our framework is designed to improve the effectiveness of ensemble defenses, and the attackers who are targeting them, through a game-theoretic approach. In this way we are finding a more optimal way to combine multiple, unique classifiers into an ensemble defense that is more robust than any one of the individual classifiers. This approach is also designed to improve upon ensembles which implement a uniform probability distribution over all defenses available to it. We show that this holds true empirically in the supplementary material.
> >
> > Our experimental scope is substantial. We provide results on 2 different image datasets utilizing 5 state of the art defenses architectures (with 11 unique classifiers studied in the GaME results alone) and 4 state of the art attacks, 2 of which are new contributions developed in the paper.
> >
> > **Comment 8:** “Regarding your experiments, you did not mention to the attackers used in $A’$”
> >
> > All attacks used are listed in the supplementary material, including their hyperparameters. One of the goals of our paper was to perform a comprehensive analysis of SOTA attack methods. This was to create the strongest possible adversary for the defender to optimize its mixed Nash strategy against. We studied APGD, MIM, MIME, and AE-SAGA. APGD has been shown to be a SOTA attack method that outperforms attacks like PGD on single model attacks. Additionally, as shown in the paper, MIME greatly out-performs MIM and APGD when attacking random transform defenses. Lastly, AE-SAGA was developed to specifically target the ensemble defense structure we created, and achieves SOTA results that out-perform MIME and APGD on both CIFAR-10 and Tiny ImageNet (as seen in the supplementary material).

---

> > ### Author Response · Authors · 2022-12-09
> > **Response to Further Comments from Reviewer rrfN (Part 3)**
> >
> > **Comment 9:** “Additionally, for the robust accuracies in Figure 2, what attack is using?”
> >
> > In section 6 we note “Here, and throughout the paper, we measure the robustness of each ensemble as the lowest robust accuracy achieved by any single attack in our study.”. We choose this methodology as the robustness of any defense is determined by the worst case scenario. For visual clarity, and to maintain the page limit, we do not list the best attack against each ensemble and each defense in figure 2. However, all of this information will be available in a public github repository upon the publication of the paper.
> >
> > **Comment 10:** ”I believe that you should report the robust accuracies w.r.t. many attacks such as PGD-x, Auto-Attack, and so on, especially the ones not in $A’$ to demonstrate the generalization ability of your approach.”
> >
> > We did not include attacks like PGD or MIM in our study as they have been shown to be less effective than the likes of APGD and MIME. Providing such information as the reviewer suggests is simply not useful as PGD is NO the most effective attack from both a pure transferability perspective as well as a single model white-box attack perspective. For reference AGPD outperforms PGD, see the following paper [B]. For PGD being outperformed in terms of transferability style attacks, see [C]. In our analysis we use the best SOTA attacks and do not include ones that have already been supplanted such as FGSM etc. The above comment demonstrates a fundamental misunderstanding about how effective attacks work for transfer and multi-model scenarios as PGD has already been surpassed.
> >
> > **Comment 11:** “Finally, I am keen on my current scores. This is because the contributions regarding MIME and AE-SAGA are incremental and the above reasons for the main contribution in Section 5.”
> >
> > We respectfully but strongly disagree with the above conclusion. Stating MIME and AE-SAGA are incremental is simply untrue. MIME breaks a SOTA defense (TIT) that was published in 2020 in CVPR and up to this point had never been successfully attacked. MIME also works well on the original non-derivative version of the randomized defense BaRT which was also a defense published recently in CVPR. As for AE-SAGA , it is a necessary and innovative development in multi-defense based attacks. Without AE-SAGA the attacker would perform far worse (we can back this up experimental if required by rerunning all experiments without AE-SAGA as proof, although this would be superfluous). Overall, we have clearly demonstrated the merit and novelty of these attacks through rigorous experimentation in our paper.
> >
> > **We have repeatedly explained both the merit and novelty of our contributions to the reviewer. We have clarified multiple points in the paper that the reviewer was confused on (and will provide edits in the final version of the paper as well for further clarification). Simply stating a technique is incremental without justification (as the reviewer has done above) is not a grounds for rejection. Hence, we kindly ask the reviewer, once again, to reconsider their evaluation of our paper in the interest of fairness. Thank you.**
> >
> > References:
> >
> > [A] "On the robustness of vision transformers to adversarial examples." In Proceedings of the IEEE/CVF International Conference on Computer Vision, pp. 7838–7847, 2021b.
> >
> > [B] "Reliable evaluation of adversarial robustness with an ensemble of diverse parameter-free attacks." In International conference on machine learning, pp. 2206– 2216. PMLR, 2020.
> >
> > [C] "Beware the black box: On the robustness of recent defenses to adversarial examples." Entropy, 23(10):1359, 2021a.

---

### Official Review · Reviewer_hEcL · 2022-10-26

**Confidence:** 3
**Clarity, Quality, Novelty And Reproducibility:** This paper is clear. The quality is r…
**Correctness:** 3
**Technical Novelty And Significance:** 2
**Empirical Novelty And Significance:** 3
**Recommendation:** 6

**Strength And Weaknesses:**

# Strength
The studies topic is an interesting and practical one. According to my knowledge, such topic has not been discussed in the past. The theoretical results seems sufficient to me. However, I have not checked its every detail.

# Weakness
The empirical verification is not very convincing nor sufficient. According to the theory established by this work, I expect the author to construct a state-of-the-art defense model by randomly choosing the prediction model from an ensemble of models.

**Summary Of The Paper:**

This paper studies how adversarial examples crafted for one defensive model transfer on other models, and correspondingly how random defensive models help adversarial robustness.

**Summary Of The Review:**

As is.

---

> ### Author Response · Authors · 2022-11-08
> **Response to Reviewer hEcL**
>
> We thank the reviewer for their feedback and comments.
>
> **Comment 1:** “The empirical verification is not very convincing nor sufficient. According to the theory established by this work, I expect the author to construct a state-of-the-art defense model by randomly choosing the prediction model from an ensemble of models.”
>
> At this time we have an additional study on the effects of n on the GaME generated ensemble will be included in the supplemental material. These experimental results show that we are able to build a state-of-the-art defense (using the same component defenses and same voting techniques) that achieves better performance than other state-of-the-art defenses. For instance, we are able to achieve a robust accuracy of 63.5% on CIFAR-10 which outperforms the best single model defense, FAT ResNet, which has a robust accuracy of 50%. We are further able to improve upon the clean accuracy of all of the CIFAR-10 defenses except for BaRT-1, achieving a clean accuracy of 96.2% compared to BaRT-1’s 98.4% clean and 3.18% robust accuracy.
>
> On Tiny ImageNet we are able to produce similarly SOTA results, achieving a robust accuracy of 44.5% along with a clean accuracy of 72.6%. This outperforms the most robust defense, BaRT-5 with robust accuracy 14.42 and clean accuracy 57.63%. It additionally has a higher clean accuracy than any of the single models other than The BiT-ViT Trash is Treasure defense which has 76.97% clean accuracy.
>
> **Concluding Remarks:** Based on the above reasoning we respectfully ask the reviewer to consider increasing their score.

---

### Official Review · Reviewer_ZYXb · 2022-10-26

**Confidence:** 3
**Correctness:** 4
**Technical Novelty And Significance:** 3
**Empirical Novelty And Significance:** 3
**Recommendation:** 6

**Clarity, Quality, Novelty And Reproducibility:**

The descriptions of the method are clear and in general the math and experiments seem correct. I did not closely inspect all the experimental details. While the attacks presented are only slight modifications of existing attacks, the game-theoretical analysis seems quite original.

**Strength And Weaknesses:**

In general, I think the idea of the paper is quite interesting and the theory and experiments are explained clearly. As far as I know, nobody has explored an optimal strategy for randomizing over multiple defenses by using game theory. The experiments seem to support the theoretical analysis and there is a nice result that the randomized strategy is more robust than any single of its members. The connection between the game theoretic analysis and transferability of adversarial examples is also quite interesting.

I think that the main weakness of the paper is that the presentation could be improved. I found that a lot of the paper seemed to focus too much on just the methods, mathematical analysis, and algorithms, and not enough on the motivation for them. This meant that I didn't appreciate the ideas behind the paper until I had more fully internalized the methodology sections. I worry that readers will just read the title, abstract, or introduction and not appreciate all the contributions.

In particular, I think there might be too much emphasis on the parts of the paper dealing with transferability of adversarial examples. This idea has been explored a lot in the past and a casual reader might assume that the paper is just another study on transferability. Instead, I think the paper might be more impactful if the authors emphasize the game-theoretic analysis and the idea of coming up with better randomized defenses. A title that might be more attention-grabbing could be something like "Optimal Randomized Adversarial Defenses via Game-Theoretic Analysis of Attack Transferability." If the authors focus on the goal of picking a good randomized defense and use the analysis of transferability more as an argument to justify that goal rather than as a primary contribution, I think more people will find the paper novel and interesting. If more space is needed to add more motivation, I think some of the details of the new attacks could be moved to the appendix since I think the game-theory ideas and results are more interesting than the attacks. But I'm just one person so maybe others will think the attack/transfer part of the paper is more interesting!

Besides the presentation, one objection to the method in the paper (and randomized defenses in general) is that a defender could just re-submit the same adversarial example until the defender randomly picks the vulnerable model. For instance, say an attacker is trying to post offensive pictures by fooling a content-filtering system on social media. Then, they could just resubmit the pictures until the content-filtering system randomly picks a vulnerable classifier. It would be good to discuss this limitation somewhere in the paper.

Smaller issues/suggestions
 * On page 9, you say "For CIFAR-10 instance r* = .573, meaning the worst expected performance of the ensemble against
these attacks is to get a robust accuracy of 57.5%." Shouldn't it be 57.3%? Also, seems like there is a typo and it should be "For CIFAR-10 for instance" or just "For CIFAR-10".
 * This seems like a relevant paper to cite, although your approach is definitely quite different: Balcan et al. Nash Equilibria and Pitfalls of Adversarial Training in Adversarial Robustness Games. (Also I realize that it came out after you submitted, so use your judgement as to whether you think it should be cited.)

**Summary Of The Paper:**

This paper provides a game-theoretic analysis of adversarial robustness. In the adversarial examples game the authors propose, attackers choose an attack method (including both the optimization method and the defense to attack) while defenders choose a defense. This game is connected to the study of transferability of adversarial examples across diverse attacks. With respect to transferability, often attacks which are strong against one attack produce adversarial examples that do not transfer well to another defense. The authors then show that one can determine an optimal randomized strategy for attacker and defender by considering a matrix game where the payoffs are the success rates of how well each attack transfers to each defense (the values called $r_{d, a}$ in the paper). An optimal defense strategy is computed for CIFAR-10 and Tiny-ImageNet given several defenses and it is shown that it has higher robust accuracy than any single one of the defenses.

**Summary Of The Review:**

Overall, I found the paper to be an interesting analysis of adversarial transferability and its connection to randomized defenses via game theory. I would encourage the authors to work on the presentation of their ideas so that others can more easily appreciate their results.

---

> ### Author Response · Authors · 2022-11-08
> **Response to Reviewer ZXYb**
>
> **Comment 1:** “In particular, I think there might be too much emphasis on the parts of the paper dealing with transferability of adversarial examples. This idea has been explored a lot in the past and a casual reader might assume that the paper is just another study on transferability.”
>
> We do agree with the reviewer, however, that a casual reader may disregard the paper as simply a transferability study, thus we will adjust the paper to highlight the game theoretic results more and, in turn, use the transferability results as motivation.
>
> **Comment 2:** “I think that the main weakness of the paper is that the presentation could be improved. I found that a lot of the paper seemed to focus too much on just the methods, mathematical analysis, and algorithms, and not enough on the motivation for them.”
>
> We agree and will adjust future revisions to ensure that the motivation of each section is clear and understandable.
>
> **Comment 3:** “This seems like a relevant paper to cite, although your approach is definitely quite different: Balcan et al. Nash Equilibria and Pitfalls of Adversarial Training in Adversarial Robustness Games.”
>
> After reading this paper we agree with the reviewer that it is related to our work and thus necessary to include as a citation.
>
> **Comment 4:** “On page 9, you say "For CIFAR-10 instance r* = .573, meaning the worst expected performance of the ensemble against these attacks is to get a robust accuracy of 57.5%." Shouldn't it be 57.3%? Also, seems like there is a typo and it should be "For CIFAR-10 for instance" or just "For CIFAR-10".”
>
> We appreciate the reviewer’s efforts in finding these mistakes and will be sure to fix these errors along with any others that may have occurred.
>
> **Concluding Response:** We greatly appreciate the reviewer’s feedback and will make the changes they suggest.

---

> ### Comment · Reviewer_ZYXb · 2022-12-12
> **Lowered my score**
>
> Due to concerns raised about novelty by the other reviewers who are more familiar with the area, I am lowering my score.

---

> > ### Author Response · Authors · 2022-12-14
> > **Response to Decreased Score from Reviewer ZYXb**
> >
> > **Comment 1:** “Due to concerns raised about novelty by the other reviewers who are more familiar with the area, I am lowering my score.”
> >
> > We would like to point the reviewer towards our responses to reviewer “once” and reviewer “rrfN” where we clarify in detail concerns about their novelty. It is important to note for reviewer “once” they have not responded to our clarifications meaning they have either accepted or do not have any valid remaining reason to dispute our novelty claims. For reviewer “rrfN” they have actually raised their score significantly after discussing further with us. If you have any specific concerns about the novelty we would also be happy to discuss them here. In light of this, would you be willing to bring up your concerns or consider changing your score? Thank you for your consideration.

---

### Decision · Program_Chairs · 2023-01-20

**Decision:**

Reject

**Justification For Why Not Higher Score:**

Not enough support from the reviewers.

**Justification For Why Not Lower Score:**

N/A

**Metareview: Summary, Strengths And Weaknesses:**

The paper considers a set of recent defenses against adversarial attacks on the image domain and uses existing attacks or develops them to capture a game-theoretic interaction between the attacks and defenses. To obtain the utility functions, they evaluate the effectiveness of different attacks on different defenses, thereby conducting a transferability study. Eventually, they consider a mixed strategy over a single-classifier or a less-than-n-ensemble selection for the defender by formulating the game as a zero-sum game and solving for a Nash-equilibria in this game. They show that this is more effective than using single defenses.

All reviewers appreciated the game-theoretic perspective on multiple attacks and defenses in adversarial robustness. The original recommendations had high variance, ZYXb and hEcL being positive; while rrfN and once recommending rejection. The main concerns expressed by rrfN and once were related to the novelty of the work (in particular the authors missing an important citation mentioned by once), as well as some aspects of the writing of the paper as raised by rrfN. Through discussion with the authors as well as the revision of the paper, some of these concerns were addressed, and rrfN increased their recommendation from 3 to 5. On the other hand, ZYXb agreed with the novelty concerns and decreased their recommendation from 8 to 6.

While this paper is quite promising in its contributions, this AC judges that the support for this paper was too low to recommend acceptance at ICLR. The extensive discussions can yield to significant improvement in the writeup, and the authors are encouraged to take them in consideration and resubmit. In particular, the revision should more properly contrast its contributions with the citation [1] from reviewer "once".